# Model Agnostic Conditioning of Boltzmann Generators for Peptide Cyclization

## Abstract

Macrocyclic peptides offer strong therapeutic potential due to their enhanced binding affinity and protease resistance, but their design remains a challenge due to limited structural data and tools that address only a narrow set of cyclization chemistries. Moreover, existing models are built to only consider ground state or mean conformations, rather than conformational ensembles that more accurately describes peptides. We introduce CYCLOPS (a Cyclic Loss for the Optimization of Peptide Structures), a model-agnostic framework that conditions Boltzmann generators to sample valid cyclic conformations—without retraining. To overcome the scarcity of cyclic peptide data, we reformulate the design problem in terms of conditional sampling over linear peptide structures via chemically informed loss functions. CYCLOPS encompasses 18 possible inter-amino acid crosslinks enabled by 6 diverse chemical reactions, and is readily extensible to many more. It leverages tetrahedral geometry constraints, using six interatomic distances to define a kernel density-estimated joint distribution from MD simulations. We demonstrate CYCLOPS's versatility via two distinct generative models: a modified Sequential Boltzmann Generator (SBG) (Tan et al., 2025a) and the Equivariant Normalizing flow (ECNF) of Klein & Noé (2024). In both settings, CYCLOPS successfully biases the Boltzmann distribution toward chemically plausible macrocycles.

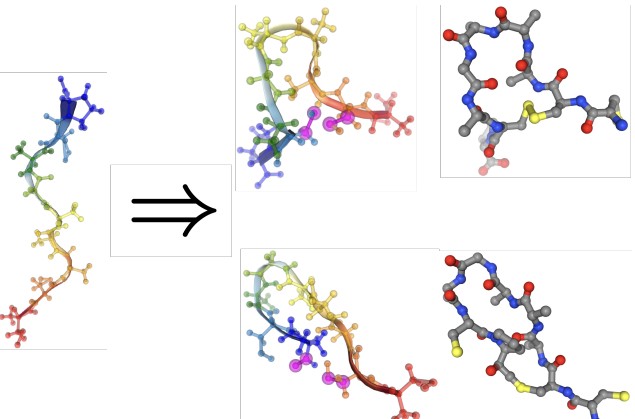

Figure 1: Cyclization of a linear peptide at two different stapling sites performed through CYCLOPS conditioning of a Boltzmann Generator.

## 1 Introduction

Cyclic peptides offer therapeutic advantages including protease resistance, enhanced binding affinity, and, in some cases, improved membrane permeability, through intramolecular crosslinking (Craik, 2006; Zorzi et al., 2017; Hayes et al., 2021; Mizuno-Kaneko et al., 2023; Ji et al., 2024). Recent generative approaches for cyclic peptide design (Li et al., 2025; Rettie et al., 2025; Zhou

et al., 2025; Zhu et al., 2025) face several challenges: they only exploit a limited set of cyclization chemistries (Bechtler & Lamers, 2021), do not fully account for conformational flexibility and complex linkage geometric constraints, consider average conformations rather than the conformational ensembles peptides adopt in solution (Huang & Nau, 2003), or require retraining to perform their conditioning (Jiang et al., 2025). Since conformational dynamics govern binding (Buch et al., 2011), and peptides exhibit more disorder than proteins (Wang et al., 2022b; Ho & Dill, 2006), accounting for the peptide's conformational Boltzmann distribution may be advantageous. Boltzmann generators (BGs) constitute a class of generative model which learn to sample from conformational ensembles by mapping noise to approximate data points. BGs fall into two categories: discrete normalizing flows (DNFs), which apply a fixed number of invertible transforms

$$f_\theta^{-1}(x_0) = f_{1,\theta_1}^{-1} \circ \cdots \circ f_{N,\theta_N}^{-1}(x_0) \tag{1}$$

and continuous normalizing flows (CNFs), which integrate a neural ordinary differential equation

$$\vec{v}_\theta^{(t)}(x_t) = dx/dt \tag{2}$$

over $t \in [0, 1]$, with $x_0 \sim \mathcal{N}(\vec{0}, \mathbf{I})$, to arrive at a final sample (Ho et al., 2019; Chen et al., 2018). Both DNFs and CNFs have shown promise for efficient conformational Boltzmann sampling (Tan et al., 2025a;b; Klein & Noé, 2024). However, conditional Boltzmann-generator-based peptide design remains largely unexplored.

**Our contributions:** This work introduces a Cyclic Loss for the Optimization of Peptide Structures (CYCLOPS). **(1)** CYCLOPS is the first framework to condition Boltzmann generators for cyclic peptide design, generating all-atom cyclic conformations by conditioning the Boltzmann ensemble. **(2)** CYCLOPS overcomes limited cyclic peptide data by leveraging available linear peptide MD simulations with chemically informed geometric constraints on 4 canonical atoms shared with the crosslinks, incorporating 18 cyclization strategies via KDE-fitted tetrahedral conditioning from small "toy" MD simulations. **(3)** CYCLOPS conditions any Boltzmann generator architecture *without retraining*—demonstrated on both DNF and CNF models via latent space simulated annealing and loss-based flow guidance, respectively.

## 2 PROBLEM FORMULATION

A significant bottleneck in applying machine learning to science is the availability of large, high-fidelity, and well-balanced datasets. This is also true of cyclic peptide design. Therefore, an essential question is how one can identify cyclizations of a linear chain that do not perturb its ensemble properties enough to affect its binding to a particular protein target. Moreover, can this be done without additional cyclic-peptide MD simulations or per-cyclization retraining? This may be reframed as a problem of statistical conditioning. First, we consider what a peptide's linear conformations reveal about (1) its binding and (2) its possible cyclizations. (1) is well-studied in the case of structured targets. It has long been known that a protein's structure is linked to its function. Hence, conformational motifs that are preserved across most of the conformational ensemble of the bound state play a crucial role in the binding interactions to a protein target. In fact, motif-constrained design has become a standard approach for developing protein binders (Yim et al., 2024; Ingraham et al., 2023; Trippe et al., 2022; Wang et al., 2022a). Yet (2) remains largely underexplored.

Consider the random variables $S$, a possible conformation of a linear amino acid chain, and $S_\kappa$, a possible cyclic conformation in any possible linkage, both implicitly conditioned on a particular initial sequence of amino acids. Of course, the distribution of $S_\kappa$ is not completely knowable given the probability density function of $S$ alone, since the chain never truly exists in a cyclic conformation; at no point do any of the bonds involved exist. Therefore, we approximate this as a problem of statistical conditioning.

Intuitively, nearly cyclic conformations—structures which almost satisfy typical bonding constraints—appearing with significant probability density suggest that the linear chain may be amenable to a given cyclization. Let $C$ be the event the chain is approximately cyclic, i.e. the constraints of a particular linkage are almost satisfied. The problem of cyclic peptide design then becomes sampling from valid linear conformations conditioned on both binding feasibility and cyclizability, as shown in the inner purple region of Fig. 2. We therefore seek to modulate the landscape of $S$ to increase the probability of sampling approximately cyclic peptides, ideally whilst preserving

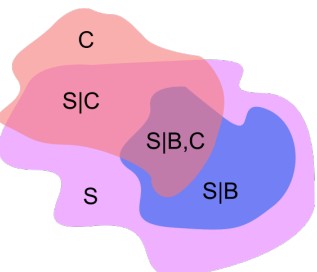

Figure 2: Schematic representation of peptide conformational spaces showing linear Boltzmann space ($S$), cyclic space ($C$), and bound ($B$) regions, with $S|B, C$ representing the desired sampling target. Note that $S|B$ is equivalent to $B$, because a state can only be bound if it is already a valid conformation of the chain. Similarly, it is trivial to arrive at cyclizations which completely perturb a linear chain and whose constraints are never approximately satisfied. Thus $C$ is not a subset of $S$.

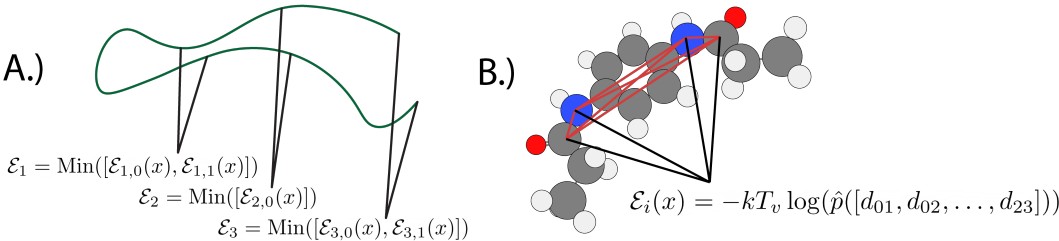

Figure 3: Schematic of the CYCLOPS framework: **(A)** Automatic identification of cyclization sites on peptide chains, and **(B)** Computation of cyclization probability via tetrahedral KDE fitted to toy model simulations and evaluated on four shared atoms. Resonant losses between chemically equivalent atoms within the same amino acids are set to their hard minimum prior to reweighting.

the dynamics of a binding region or motif. To sufficiently limit the scope of this paper, we shall principally consider $S|C$, as conditioning on binding is somewhat well studied.

## 3   METHODS

We consider structure conditioning in terms of a constraint defining function (CDF), whose minima represent samples which closely approximate the geometric constraints of cyclicality. The Boltzmann micro-canonical ensemble yields an ideal CDF:

$$\mathcal{E}_i(x) = -k_B T \log p(x|C_i) \tag{3}$$

where $C_i$ is the event that the chain is subject to the constraints of a particular cyclization and $x$ is the peptide's conformation.[1] Our insight is that a chain's degree of cyclicality under a given cyclization chemistry can be characterized by four atoms it shares with a toy model of the linkage in question (see Appendix A.5 for the chemical composition of these models); these four atoms form a tetrahedron whose side lengths capture the spatial relationships required for cyclization, given a sufficiently small linkage.[2] The toy models of each cyclization may then be simulated via Langevin molecular dynamics, but how can the resulting data points be used to approximate $p(x|C_i)$? One approach is to use generative models, which have become increasingly popular for density estimation (Ho et al., 2019; Dinh et al., 2016). However, these are prohibitively slow for our conditioning. As such, we employ Kernel Density Estimation (KDE) (Rosenblatt, 1956; Parzen, 1962), to convert our data to approximate distributions (see Appendix A.1 for details).

---

[1]Jiang et al. (2025) consider a similar formulation, though $p(x)$ is implicitly learned during training. This makes the framework not model agnostic and represents a significant limitation.

[2]Sufficiently small can be defined as small enough that atoms outside of this tetrahedron do not interact with the linkage's virtual atoms. This represents a significant limitation.

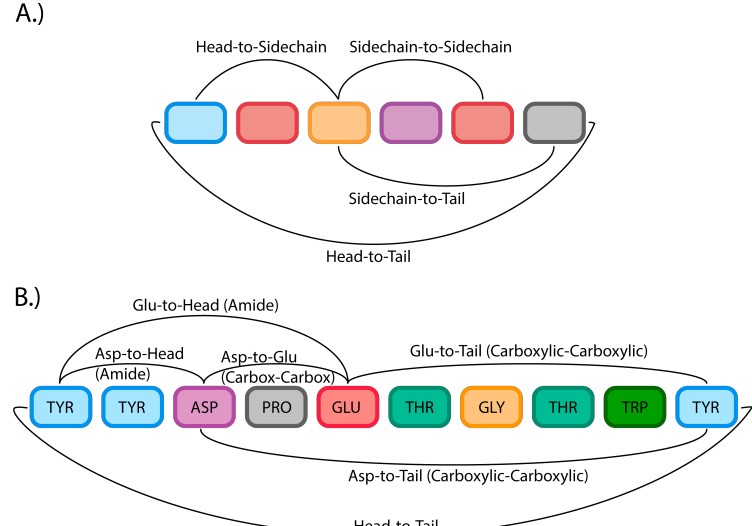

Figure 4: **A.)** A diagram illustrating some of the chemical permutations possible for peptide cycliza-tions. **B.)** Some of the possible cyclizations for the chignolin peptide (Suenaga et al., 2007), based on some of the chemistries from Bechtler & Lamers (2021).

However, real peptides often admit multiple cyclizations, as illustrated in Fig. 4. We therefore desire an *expected* cyclic loss

$$\mathcal{L}_{\text{cyc}} = \sum_i^N P(C_i|x)\mathcal{E}_i(x) \tag{4}$$

where $N$ is our number of allowed cyclizations. We make the maximum entropy assumption that

$$P(C_i|x) \propto \exp(-\alpha\mathcal{E}_i(x)) : \alpha > 0 \tag{5}$$

(see Appendix A.2 for a proof), which has the convenient property of becoming a soft minimum if $\alpha >> 1/k_B T$. We may now sample from valid cyclic conformations, and look at the frequency with which various cyclizations are chosen to determine the best ways to cyclize a given peptide whilst minimally perturbing its native Boltzmann distribution. The functionality of the CYCLOPS framework is summarized in Fig. 3.

## 4 RESULTS

To test CYCLOPS DNF conditioning, we perform simulated annealing of latent space vectors us-ing the Sequential Boltzmann Generator (SBG) Tarflow architecture (Zhai et al., 2024) of Tan et al. (2025a), without their post-generation Langevin annealing (see Appendix A.3 for details). Simu-lated annealing of the CYCLOPS loss produces valid cyclic structures from a Boltzmann generator trained only on linear peptide data, as demonstrated across three distinct sequences (Fig. 4).

The conditional samples correspond to valid 3D structures when cyclization atoms are added with correct bond topology. Only the added atoms are relaxed using force field gradient descent [3]. As seen in Fig. 7, DNF-simulated annealing appears to explore prior space since the distribution of finally chosen cyclizations differs from that of the smallest loss before annealing. In the case of PVAAKKIKW and CCAAAGACP, this involves changing the most frequent smallest loss. This is also apparent on a sample-wise level in Fig. 6, which shows the pre- and post-conditioning structures of the samples from Fig. 4, which differ substantially. All together, this suggests that the loss minimization observed in Fig. 11 results from genuine latent space exploration rather than simply refining the initially generated structure.

We test CYCLOPS's ability to condition CNFs using the Equivariant Normalizing Flow++ (ECNF++) of Klein & Noé (2024); Tan et al. (2025a). We generate samples, filter out mixed chirality

---

[3]Good initial guesses must be provided, however, as gradient descent is prone to kinetic traps.

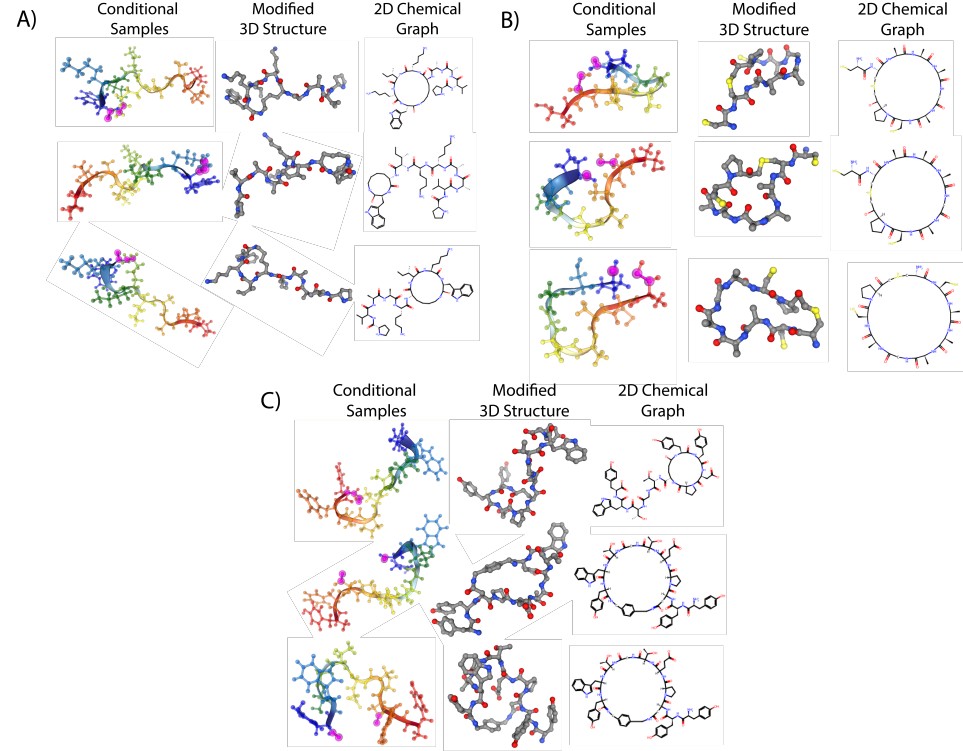

Figure 5: Conditional samples from CYCLOPS simulated annealing using SBG for peptides **(A)** PVAAKKIKW, **(B)** CCAAAGACP, and **(C)** Chignolin: (*left*) generated conformations with cyclization sites in pink, (*center*) 3D models with added cyclization atoms, (*right*) 2D skeletal representations.

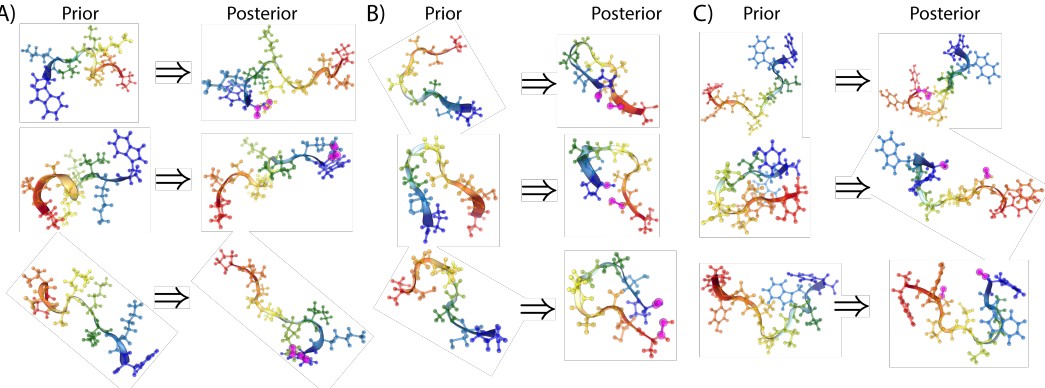

Figure 6: Conformational changes before and after $\mathcal{L}_{\text{cyc}}$ simulated annealing demonstrate latent space exploration for sequences: **(A)** PVAAKKIKW, **(B)** CCAAAGACP, **(C)** Chignolin.

conformations, and select those with lowest $\mathcal{L}_{\text{cyc}}$. Results in Fig. 8 show guided flow conditioning plus filtering (*left*) versus just filtering (*right*). Low $\mathcal{L}_{\text{cyc}}$ corresponds to cyclic conformations, with flow guidance showing additional improvement. However, this approach yields lower sample quality, with C-terminal caps frequently separating from the molecule and bulky sidechains becoming distorted during conditioning. This suggests head-to-tail amide cyclization drives the flow out-of-distribution.

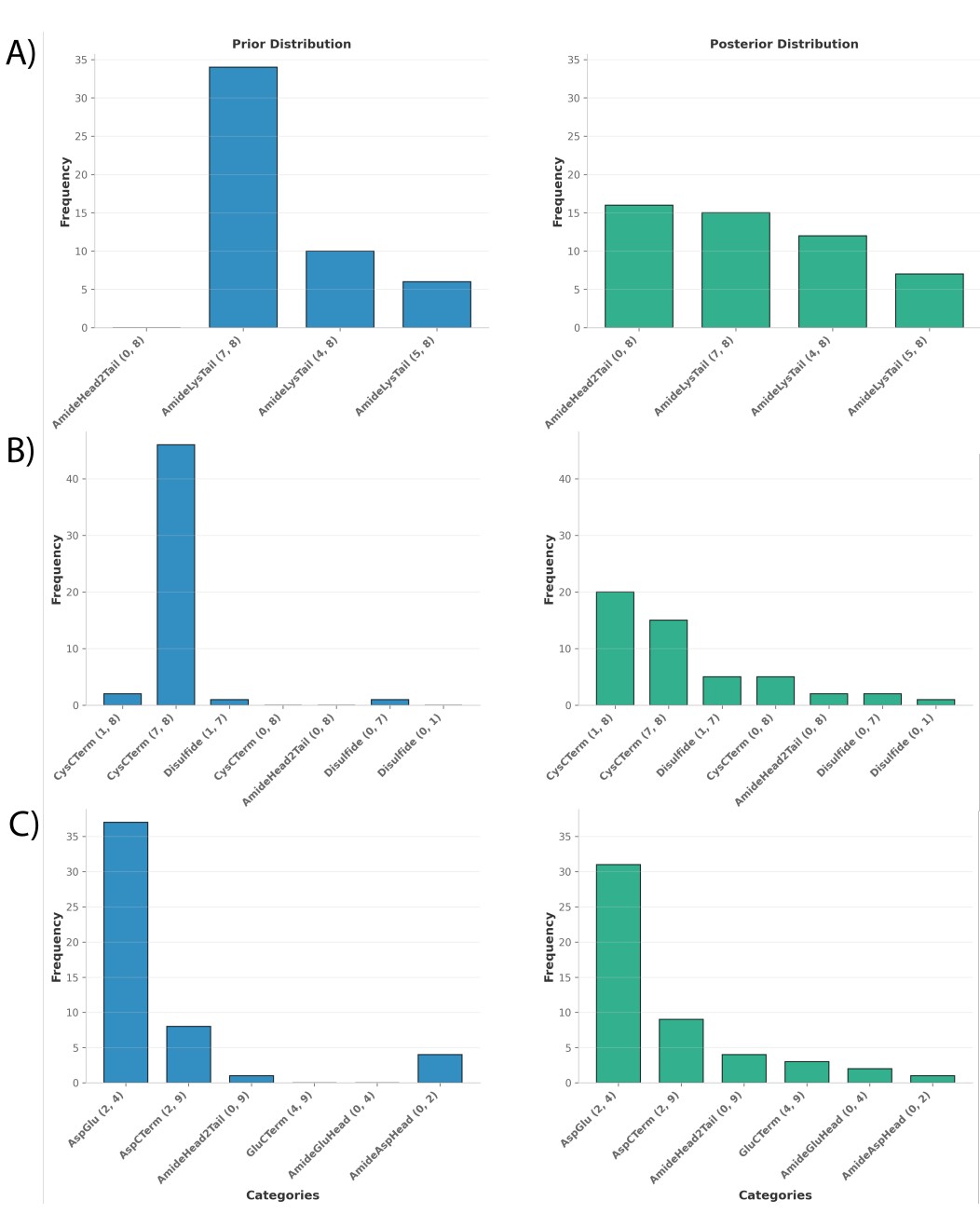

Figure 7: Prior and posterior distributions of minimum chemical losses reveal that cyclization preference shifts after simulated annealing. Histograms illustrate the lowest individual chemical loss within $\mathcal{L}_{\text{cyc}}$ before (prior) and after (posterior) simulated annealing of 50 samples for **(A)** PVAAKKIKW, **(B)** CCAAAGACP, and **(C)** Chignolin. Each cyclization is labeled by its chemistry, with bonded amino acid positions in parentheses. Simulated annealing systematically shifts the loss distributions, with the most favorable cyclization changing for peptides A and B.

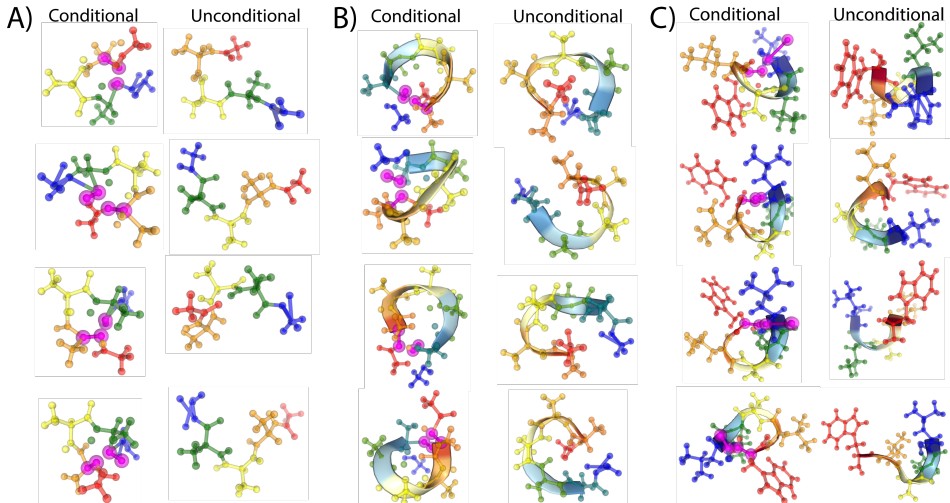

Figure 8: Guided flow conditioning vs. unconditional generation; in both cases, samples shown represent the four lowest CYCLOPS loss conformations for peptides: **(A)** AAAA, **(B)** AAAAAA, and **(C)** WLALL. Pink atoms indicate CYCLOPS tetrahedron atoms.

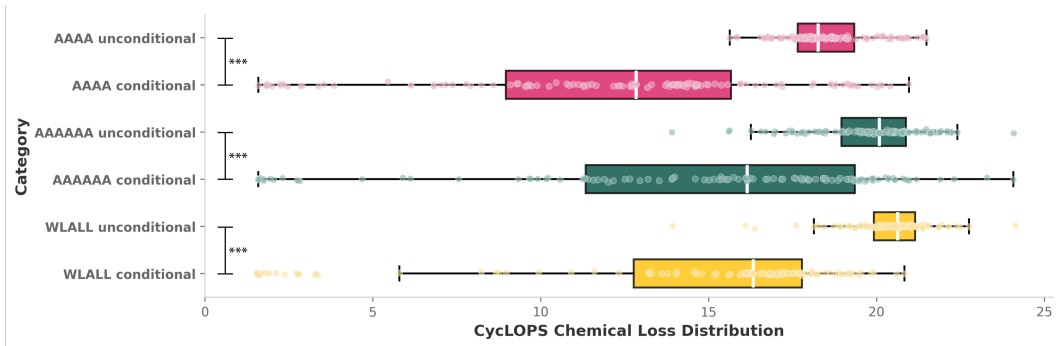

Figure 9: Box plots of CYCLOPS loss distributions for unconditioned vs. conditioned-with-guided-flow ECNF++ samples across peptide sequences. The $E(3)$-equivariant ECNF++ is prone to producing samples of the wrong-chirality. Generated samples of entirely the wrong chirality are reflected, whilst those of mixed chirality are discarded. Conditioning used loss-flow guidance ($\omega = 1.06$), significantly altered distributions (Kolmogorov–Smirnov test) and reduced mean loss (permutation test, 10,000 permutations) for all sequences ($p < 0.005$, both tests).

The statistical analysis in Fig. 9 demonstrates two key findings: **(1)** injecting our prior through guided flow conditioning significantly reduces the cyclic loss distribution across all tested peptides, and **(2)** this reduction corresponds to more cyclic (though potentially less physically plausible) structures. The conditioning succeeds in significantly reducing the distribution of CYCLOPS losses for all tested peptides, though not to the degree achieved by simulated annealing (Fig. 11). In all cases, conditioning increases the variance of the distributions and produces apparent clusters of datapoints at the lower extrema of our losses. This suggests that while the prior injection effectively drives the model toward cyclic conformations, it may also push the flow somewhat out-of-distribution, leading to structures that are more cyclic but potentially less physically realistic than those generated by the unconditional model.

## 5 CONCLUSION

We present CYCLOPS, the first model-agnostic framework for conditioning Boltzmann generators to sample cyclic peptide conformations without retraining. By reformulating cyclization as conditional sampling with tetrahedral geometric constraints derived from toy MD simulations, CYCLOPS successfully generates valid cyclic structures across both DNF and CNF architectures, with DNF-simulated annealing demonstrating superior sample quality.

CYCLOPS addresses three challenges in computational peptide design: **(1) Data scarcity**—repurposing abundant linear peptide trajectories rather than requiring expensive cyclic datasets; **(2) Model-agnostic conditioning**—enabling cyclization without architectural modifications or retraining across diverse generative models; **(3) Chemical diversity**—encompassing 18 inter-amino acid crosslinks across 6 distinct reaction chemistries, representing coverage unmatched in the literature.

The encoding of cyclization feasibility through four shared canonical atoms forming KDE-approximated tetrahedral constraints establishes a generalizable template for incorporating complex chemical knowledge into existing generative models. This ensemble-based approach captures conformational dynamics rather than static ground states, potentially revealing cyclization opportunities invisible to structure-based methods. Key limitations include small linkage chemistry assumptions and optimization parameter sensitivity (see Sec. B.1.3). Future work will include: (1) expanding cyclization chemistries through new MD-based losses, (2) integrating binding motif preservation, (3) targeting therapeutically relevant peptides (PD-1, MDM2, MDMX), and (4) conditioning sequence-transferable Boltzmann generators.

By demonstrating that complex geometric constraints can be imposed without retraining, we hope CYCLOPS serves as a foundation for new avenues for knowledge-guided molecular generation. The combination of model-agnostic conditioning, chemical diversity, and data efficiency suggests utility in therapeutic cyclic peptide discovery.

## 6 REPRODUCIBILITY STATEMENT

We provide detailed training parameters, hyperparameters, computational resources, and implementation details in Tables 1 and 2, along with specific algorithm descriptions. However, code and data are not publicly released to preserve anonymity during review and to protect ongoing research directions that build upon this foundational work, preventing potential scooping of future iterations and extensions of the project.

## 7 ETHICS STATEMENT

Our research follows ethical guidelines for computational research, uses publicly available datasets with proper attribution, and develops methods for beneficial applications in drug discovery (see Sec. A.5). The methods are intended for beneficial drug discovery research and we do not forsee negative societal implications.

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

## A  TECHNICAL DETAILS AND SUPPLEMENTARY MATERIAL

### A.1  KERNEL DENSITY ESTIMATOR IMPLEMENTATION

Kernel density estimators are defined as

$$\hat{p}(\vec{x}|\mathbf{H}) \equiv \frac{1}{n} \sum_{\vec{\delta} \in \mathcal{D}} K_{\mathbf{H}}(\vec{x} - \vec{\delta}) \tag{6}$$

and

$$K_{\mathbf{H}}(\vec{x}) \equiv \det(\mathbf{H})^{-1/2} K(\mathbf{H}^{-1/2}\vec{x}) \tag{7}$$

where $K$ is our kernel, $\mathcal{D}$ contains our simulated tetrahedral distances for the linkage in question, $\mathbf{H}$ represents a positive definite bandwidth matrix and $\det(\mathbf{M})$ denotes the determinant of matrix $\mathbf{M}$. In this instance, a kernel is a positive function which integrates to one. We allow our data to be vectors of 6 select interatomic distances which comprise a tetrahedron.

For ease of optimization, we additionally require $K$ be unimodal, spherically symmetric, centered on the origin, and smooth (Chacón & Duong, 2018), with heavy tails to ensure finite values once the log is applied and non zero gradients far from the data. Thus the Cauchy distribution

$$K(\vec{x}) = \frac{\Gamma\left(\frac{d+1}{2}\right)}{\left(\pi\left(1 + \|\vec{x}\|^2\right)\right)^{\frac{d+1}{2}}} \tag{8}$$

is ideal for the reasons it is normally difficult to work with. We fit estimators with a bandwidth of $0.64\text{Å} * \mathbf{I}$. Ideally, modern multivariate kernel smoothing techniques, like the multivariate extension of the method of Sheather & Jones (1991) proposed by Chacón & Duong (2010), should be used; due to the non-triviality of developing a Python implementation, we leave this to future work. For PyTorch compatible kernel density estimation, where loss calculations are automatically differentiable and benefit from GPU acceleration, we use the implementation provided by Kladny (2025). However, CYCLOPS should not only compute $\mathcal{L}_{\text{cyc}}(x)$, but also seamlessly handle the identification of possible cyclizations. This functionality is implemented largely through the MDTraj Python package (McGibbon et al., 2015).

### A.2  PROOF OF MAXIMUM ENTROPY REWEIGHTING

By Bayes rule:

$$P(C_i|x) \propto p(x|C_i)P(C_i) \tag{9}$$

If one has a plethora of MD trajectories for diverse chains, one could compute $P(C_i) = \sum_{s \in \mathcal{S}_s} P(s) \int P(C_i|x, s) p(x|s) dx$ where $\mathcal{S}_s$ is the set of all amino acid sequences. As there is no

such data widely available yet, we make the maximum entropy assumption that $P(C_i)$ is uniform over its support. By $\mathcal{E}_i(x) = k_B T \log p(x|C_i)$, we get

$$P(C_i|x) \propto \exp(-\mathcal{E}_i(x)/k_B T) \tag{10}$$

Note that the standard exponential maximum entropy result (Dowson & Wragg, 1973) does not apply here, since we are computing, rather than constraining, our mean. If we decouple the distribution temperature, $T_d$, from that of each $\mathcal{E}_i$, notated $T_\mathcal{E}$, we may define

$$P_\alpha(C_i|x) = \frac{\exp(-\alpha\mathcal{E}_i(x))}{\sum_{i \in \mathcal{S}_c} \exp(-\alpha\mathcal{E}_i(x))} : \alpha \equiv 1/k_B T_d \tag{11}$$

If $\alpha$ is sufficiently large, this prevents optimal structures from being an unphysical superposition of cyclizations. Hence we define

$$\text{SoftMin}_\alpha([\mathcal{E}_0, ..., \mathcal{E}_{N-1}]) \equiv \sum_{i=0}^{N-1} P_\alpha(C_i|[\mathcal{E}_0, ...\mathcal{E}_{N-1}])\mathcal{E}_i \tag{12}$$

This is tantamount to renormalization, given a set of energies, according to what their probabilities would have been if generated with temperature $T_d$. In practice, we take $k_B = 1$ and $\alpha = 3$.

### A.3 MODEL-AGNOSTIC CONDITIONING

Given our cyclic loss $\mathcal{L}_{\text{cyc}}$, how do we condition Boltzmann generators without retraining? A universal method is to optimize over model latent space, searching for conformations which minimize our CDF (Abdin & Kim, 2024; Noé et al., 2019). If the underlying network is differentiable, one could use a stochastic gradient descent-based optimizer, like ADAM (Kingma & Ba, 2014). A strategy without this requirement, however, is simulated annealing, which has already seen success in cyclic peptide design (Zhu et al., 2025). This is a probabilistic optimization algorithm inspired by annealing in metallurgy, where a material is heated and then slowly cooled to reach an energetic minimum (a highly ordered crystal structure). Here, our system's "energy" is some function of our state we seek to minimize. As illustrated in Alg. 1, simulated annealing involves sequentially updating some state $x_t$ by proposing a new state $x'$, computing the difference between their energies, and switching states as a function of some gradually cooling temperature and the change in energy. As the temperature cools, the algorithm gets more greedy, therefore modulating between exploratory and exploitative behavior as it runs its course. Thus it is not as prone to local minima as other optimization algorithms if well tuned (Press et al., 2002).

Simulated annealing is, by nature, suited to discrete optimization. We adapt this to our continuous usecase by assigning the system a velocity $r_t$ at each timestep, which then determines the radius of a hypersphere from which a proposal state is sampled from. Inspired by the relationship between temperature and mean particle velocity, we allow $r_t$ to scale with the square root of the ratio of the current to initial temperature. Since the probability of switching between states depends directly on $\Delta E/T_t$, care must be taken to pick an appropriate temperature; too large, and the system will "melt," switching between states with no regard for their energy. Too small, and the system will collapse into whatever minimum is most proximal to its initial state. We therefore set an appropriate $T_0$ based on a calculated mean absolute initial $\Delta E$, which is then scaled by a user set hyperparameter $\kappa$ that determines the starting switching probability. We then exponentially cool the simulation to zero by multiplying the previous temperature at each timestep by a constant $\lambda \lessapprox 1$ to determine its new temperature; this is so that the system is cooled sufficiently slowly, which, given enough steps, helps the algorithm find a global minimum. Our specific implementation is shown in Alg. 2. In practice, we set temperature determining $\kappa$ to 2, cooling rate $\lambda$ to 0.995, and define our objective function $E(\cdot) \equiv \mathcal{L}_{\text{cyc}}(\cdot)$. It may be advantageous to construct the objective function more cleverly, such that it considers additional desiderata. This may include binding motif preservation or OpenMM-based conformational energy.

For continuous normalizing flows, direct latent optimization can be prohibitively slow. Instead, one can guide a CNF to sample from an approximate conditional posterior. In its simplest incarnation, this takes the form of Bayes rule (Chung et al., 2022; Jiang et al., 2025):

$$\nabla_{x_t} \log p_t(x_t|C) \propto \nabla_{x_t} \log p_t(x_t) + \nabla_{x_t} \log p_t(C|x_t) \tag{13}$$

---

**Algorithm 1:** Generic Simulated Annealing

---

**Input:** Number of steps $N_{\text{step}}$, initial temperature $T_0$, cooling schedule $\text{Cool}(\cdot, \cdot)$, acceptance probability function $P_{\text{accept}}(\cdot, \cdot)$, objective function $E(\cdot)$

**Output:** Final state $x_t$ after simulated annealing

**Initialize:** Sample initial state $x_0 \sim q(\cdot)$;

$x_t \leftarrow x_0$;

$t \leftarrow 0$;

**while** $t < N_{step}$ **do**

    Generate proposal state: $x' \sim \text{Proposal}(x_t, T_t)$;

    Compute energy difference: $\Delta E \leftarrow E(x') - E(x_t)$;

    Compute acceptance probability: $p_{\text{accept}} \leftarrow P_{\text{accept}}(\Delta E, T_t)$;

    Sample $u \sim \text{Uniform}(0, 1)$;

    **if** $u \leq p_{accept}$ **then**

        $x_t \leftarrow x'$ ;                     `// Accept proposal`

    **else**

        $x_t \leftarrow x_t$ ;                     `// Reject proposal`

    **end**

    $t \leftarrow t + 1$;

    Update temperature: $T_t \leftarrow \text{Cool}(T_t, t)$;

**end**

**return** $x_t$;

---

**Algorithm 2:** Batch Simulated Annealing for Chemical Loss Optimization

---

**Input:** Number of steps $N_{\text{step}}$, batch size $B$, number of atoms $N_{\text{atom}}$, initial step size $r_0$, cooling rate $\lambda \in (0, 1]$, initial temperature scaling $\kappa$, objective function $E(\cdot)$

**Output:** Final batch of states $\mathbf{S}_{\text{final}} \in \mathbb{R}^{B \times N_{\text{atom}} \times 3}$

**Initialize temperature:**;

Sample initial batch for calibration: $\mathbf{S}_{\text{calib}} \sim \mathcal{N}(\mathbf{0}, \mathbf{I})^{B \times N_{\text{atom}} \times 3}$;

Generate proposal batch: $\mathbf{S}'_{\text{calib}} \leftarrow \mathbf{S}_{\text{calib}} + r_0 \cdot \text{UnitSphere}(B, N_{\text{atom}} \times 3)$;

Compute energy differences: $\Delta \mathbf{E}_{\text{calib}} \leftarrow E(\mathbf{S}'_{\text{calib}}) - E(\mathbf{S}_{\text{calib}})$;

Set initial temperature: $T_0 \leftarrow \kappa \cdot \text{mean}(|\Delta \mathbf{E}_{\text{calib}}|)$;

**Initialize main algorithm:**;

Sample initial batch: $\mathbf{S}_0 \sim \mathcal{N}(\mathbf{0}, \mathbf{I})^{B \times N_{\text{atom}} \times 3}$;

$\mathbf{S}_t \leftarrow \mathbf{S}_0$;

$t \leftarrow 0$;

$\epsilon \leftarrow 10^{-8}$ ;                     `// Prevent division by zero`

**while** $t < N_{step}$ **do**

    Update temperature: $T_t \leftarrow T_0 \cdot \lambda^t$;

    Update step size: $r_t \leftarrow r_0 \cdot \sqrt{T_t/T_0}$;

    **for** *each sample* $i = 1, \ldots, B$ **do**

        Generate proposal: $\mathbf{s}'_{t,i} \leftarrow \mathbf{s}_{t,i} + r_t \cdot \mathbf{u}_i$ where $\mathbf{u}_i \sim \text{UnitSphere}(N_{\text{atom}} \times 3)$;

        Compute energies: $E_{\text{old}} \leftarrow E(\mathbf{s}_{t,i})$, $E_{\text{new}} \leftarrow E(\mathbf{s}'_{t,i})$;

        Compute energy difference: $\Delta E_i \leftarrow E_{\text{new}} - E_{\text{old}}$;

        Compute acceptance probability: $p_{\text{accept},i} \leftarrow \left(1 + \exp\left(\frac{\Delta E_i}{\max(T_t, \epsilon)}\right)\right)^{-1}$;

        Sample $u_i \sim \text{Uniform}(0, 1)$;

        **if** $u_i \leq p_{accept,i}$ **then**

            $\mathbf{s}_{t+1,i} \leftarrow \mathbf{s}'_{t,i}$ ;             `// Accept proposal`

        **else**

            $\mathbf{s}_{t+1,i} \leftarrow \mathbf{s}_{t,i}$ ;             `// Reject proposal`

        **end**

    **end**

    $t \leftarrow t + 1$;

**end**

**return** $\mathbf{S}_{N_{step}}$;

---

In many instances, the connection between $x_t$ and $C$ must be learned, which can be challenging; in these cases, it is preferable to construct a differentiable CDF for the condition in question (Song et al., 2023). For denoising diffusion probabilistic models (Ho et al., 2020)–since CDFs are naturally defined over the final structure space rather than the noisy intermediate states–we can employ the following approximation

$$\hat{x}_1(x_t) \equiv \mathbb{E}[x_1|x_t] = \frac{1}{\sqrt{\bar{\varsigma}_t}}(x_t + (1 - \bar{\varsigma}_t)\nabla_{x_t} \log p(x_t)) \tag{14}$$

where $\varsigma : t \in [0,1] \to (0,1]$ is some scheduler, $\bar{\varsigma}_t \equiv \prod_{s \in \mathcal{S}_{\text{eval}}} \varsigma_s$, and $\mathcal{S}_{\text{eval}}$ is the set of discrete evaluation timesteps in $(0,1)$ (Ho et al., 2020; Komorowska et al., 2025). This has the intuitive explanation of linearly interpolating between the current state and the final state based on the present vector field of the neural ODE to arrive at an expected output. Thus, we get:

$$\nabla_{x_t} \log p(C|x_1) \approx \nabla_{x_t} \log p(C|\hat{x}_1(x_t)) = -\nabla_{x_t} \log \exp \mathcal{L}_{\text{cyc}}(\hat{x}_1(x_t)) = -\nabla_{x_t} \mathcal{L}_{\text{cyc}}(\hat{x}_1(x_t)) \tag{15}$$

where $\mathcal{L}_{\text{cyc}}$ enforces $C$ at $t = 1$. Since diffusion is effectively optimal transport flow matching (Gao et al., 2024),[4] we construct the heuristics:

$$\vec{v}_{\text{tot}}^{(t)} = \vec{v}_\theta^{(t)}(x_t) - \omega \nabla_{x_t} \mathcal{L}_{\text{cyc}}(\hat{x}_1(x_t)) \tag{16}$$

$$\hat{x}_1(x_t) = (1 - t)\vec{v}_\theta^{(t)}(x_t) + x_t \tag{17}$$

where $\omega > 0$ represents our conditioning strength; greater values will help ensure the condition is satisfied, but will contribute to lower sample quality. Additionally, by the chain rule:

$$\nabla_{x_t} \mathcal{L}_{\text{cyc}}(\hat{x}_1(x_t)) = \frac{\partial \hat{x}_1}{\partial x_t} \nabla_{\hat{x}_1} \mathcal{L}_{\text{cyc}}(\hat{x}_1) \tag{18}$$

$$\frac{\partial \hat{x}_1}{\partial x_t} = (1 - t)\frac{\partial \vec{v}_\theta^{(t)}}{\partial x_t} + \mathbf{I} \tag{19}$$

Komorowska et al. (2025) note that this comprises of an easy to compute gradient $\nabla_{x_1} \mathcal{L}_{\text{cyc}}(\hat{x}_1)$ and an expensive Jacobian matrix $\partial \vec{v}_\theta^{(t)}/\partial x_t$, which can be empirically ignored (set to $\mathbf{0}$). Thus, CYCLOPS provides a unified framework for conditioning any all-atom Boltzmann generator toward diverse cyclic peptide conformations without retraining. This can be done in all cases via simulated-annealing-based latent space optimization, or flow guidance if working with a CNF.

## A.4 BOLTZMANN GENERATOR DETAILS

Conditioning is only as good as the underlying model. As such, we begin by discussing the details of our Boltzmann generators. We approximately follow the training procedures enumerated in Tan et al. (2025a) to train our models, with the exception of the energy $\mathcal{W}_1$ distance-based early stopping and the restriction of our training set to the first 100,000 frames present in the MD simulation. This is because we do not perform importance-based annealing after generation, so our underlying model must ideally be more robust. We perform no such annealing to avoid the costs associated with SDE solvers. As substantially larger training sets are used, we reduce the maximum number of epochs depending on the number of samples. We employ the Chignolin TarFlow (Zhai et al., 2024) architecture, as used by Tan et al. (2025a), to study DNFs (termed SBG for simplicity), and ECNF++, an improved version of Klein & Noé (2024)'s TBG, to study CNFs. As large MD trajectories are very expensive to simulate, we use those provided in the literature when possible. This notably includes *ab initio* Chignolin at DFT level (Wang et al., 2023), classical MD trajectories across diverse sequences (Zhu, 2021), and alanine peptides of various lengths (Schopmans & Friederich, 2025). Training data and architecture specifics are provided in Table 1, while training implementation details and resources consumed are included in Table 2.

---

[4]This is only rigorously true if the diffusion uses DDIM sampling (Song et al., 2020) and the flow matching ODE is solved with Eulerian integration. We direct the reader to Gao et al. (2024) for more details.

Table 1: Training Data and Architecture Details

| Type | Sequence | Training Set Details | | | Params | Epochs |
|------|----------|------|------|------|--------|--------|
| | | T(K) | Size(Frames) | Data Src. | | |
| SBG | GYDPETGTWG (Chignolin) | 340 | 4.6M | Wang et al. (2023) | 114M | 80 |
| SBG | PVAAKKIKW | 300 | 480K | Zhu (2021) | 114M | 300 |
| SBG | CCAAAGACP | 300 | 480K | Zhu (2021) | 114M | 300 |
| ECNF++ | AAAA | 300 | 10M | Schopmans & Friederich (2025) | 2.3M | 10 |
| ECNF++ | AAAAAA | 300 | 10M | Schopmans & Friederich (2025) | 2.3M | 10 |
| ECNF++ | WLALL | 300 | 100K | Scherer et al. (2015) | 2.3M | 1000 |

Table 2: Training Details and Computational Resources

| Type | Sequence | Batch Size (cumulative) | Number of GPUs | Total GPU Hours |
|------|----------|------|------|------|
| SBG | GYDPETGTWG (Chignolin) | 2048 | 8 | 351.2 |
| SBG | PVAAKKIKW | 2048 | 8 | 304.6 |
| SBG | CCAAAGACP | 1024 | 4 | 241.2 |
| ECNF++ | AAAA | 2048 | 4 | 35.9 |
| ECNF++ | AAAAAA | 1024 | 4 | 81.2 |
| ECNF++ | WLALL | 1024 | 8 | 165.6 |

## A.5 Linkage Simulation and Fitting Details

CYCLOPS relies on a knowledge of the approximate joint distributions of the edge lengths of a tetrahedron which encodes the geometry of a given linkage. We must therefore simulate the toy models of each of these linkages. In all cases, this was performed with the OpenMM python package (Eastman et al., 2013) with initial configurations generated via PACKMOL Martínez et al. (2009). Forces were generated via the Amberff14SB forcefield (Maier et al., 2015) and the TIP3 water model (Jorgensen et al., 1983). Molecules were parameterized via SMIRNOFF (Mobley et al., 2018) using OpenFF (Qiu et al., 2021; Boothroyd et al., 2023), with volume calculations handled by CCTK (Wagen & Kwan, 2020). Simulation code is adapted from Wagen (2024). Each small molecule was simulated for 5 ns on one NVidia A100-80GB VRAM GPU over 10 seeds with a step size of 1 fs at 300K and with a friction coefficient of 1 $ps^{-1}$. The first nanosecond of each simulation was discarded for equilibration, leaving a total effective simulation time of 40 ns. The molecule used to model each linkage constraint is shown in Fig. 10, along with the atoms used to define the tetrahedra on which the KDEs are fit. Given the small size of each system, we observe that this was quite computationally inexpensive.

The joint distributions of the six distances defined by the four specified atoms (forming the vertexes of the tetrahedron) were then fit with a KDE with a bandwidth of $0.64\text{Å} * \mathbf{I}$. Ideally, modern multivariate kernel smoothing techniques, like the multivariate extension of the method of Sheather & Jones (1991) proposed by Chacón & Duong (2010), should be used; due to the non-triviality of developing a Python implementation, we leave this to future work. For PyTorch compatible kernel density estimation, where loss calculations are automatically differentiable and benefit from GPU acceleration, we use the implementation provided by Kladny (2025). However, CYCLOPS should not only compute $\mathcal{L}_{\text{cyc}}$, but also seamlessly handle the identification of possible cyclizations. This functionality is implemented largely through the MDTraj Python package (McGibbon et al., 2015).

## B Additional Experimental Results

### B.1 Simulated Annealing

#### B.1.1 Statistical Validation of Loss Reduction

We examine not only individual samples produced by DNF CYCLOPS conditioning, but also the effects this has on sample distributions. As shown in Fig. 11, for all peptides used, CYCLOPS annealing guides samples toward statistically significant lower values of $\mathcal{L}_{\text{cyc}}$. It should be noted

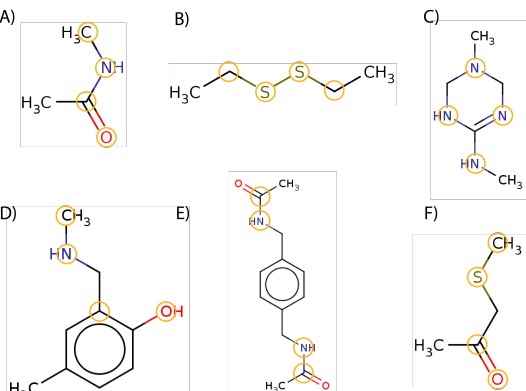

Figure 10: Schematic of the toy models used to fit KDEs for each cyclization chemistry. Shown are: **(A)** amide bond cyclization model, **(B)** disulfide bond cyclization model, **(C)** lysine–arginine cyclization model, **(D)** lysine–tyrosine cyclization model, **(E)** carboxyl–carboxyl cyclization model, and **(F)** cystine–carboxyl cyclization model. The atoms used to fit the tetrahedral KDE are highlighted in yellow. Note that these need not correspond to atoms of the same species after the chemistry is applied, highlighting the versatility of CYCLOPS. For instance, the nitrogens of **(E)** initially corresponded to carboxyl oxygens in the linear peptide, formed of canonical amino acids.

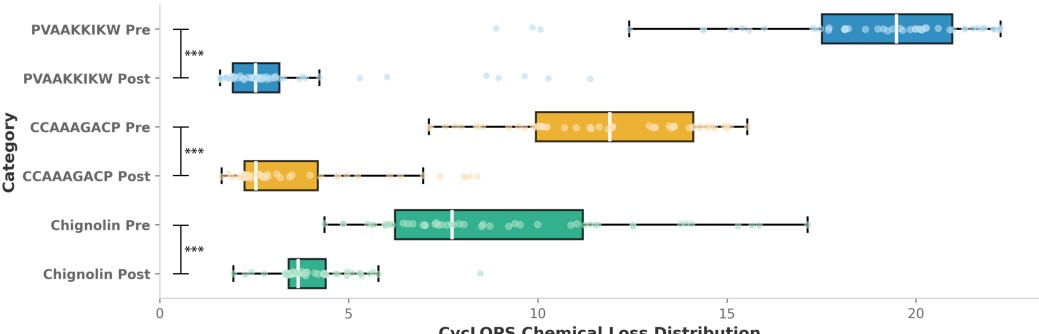

Figure 11: Box and whisker plot of the distribution of the CYCLOPS cyclic losses of 50 random samples before and after simulated annealing for representative peptides. In all cases, simulated annealing significantly changes the distribution ($p < 0.001$ represented by ***, two-sample Kolmogorov–Smirnov test). All mean-to-mean differences between pre- and post-annealing distributions are significant up to 3 decimal places under a permutation test with 10,000 random permutations.

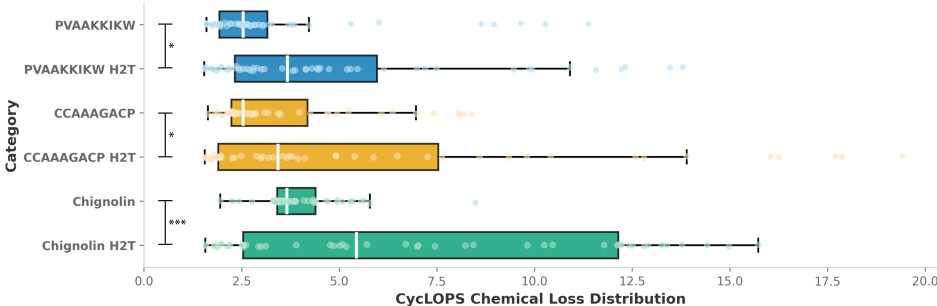

Figure 12: Box plots of CYCLOPS cyclic loss distributions for 50 random samples of representative peptides after simulated annealing, comparing all-cyclizations (samples from Fig. 11) versus head-to-tail amide bonds only (H2T). H2T distributions show higher mean and median values, greater variance, and are statistically distinct from unrestricted cyclization (* $p < 0.05$, *** $p < 0.001$, Kolmogorov–Smirnov test). Mean differences are also significant under permutation test with 10,000 random permutations ($p < 0.05$).

that most of these samples do not correspond to valid cyclic peptides: in practice, we find that only about 10% of samples in each case do not have clashes. This sampling paradigm—generating many candidates and selecting the best—is ubiquitous across protein generative AI, where the power lies not in perfect individual samples but in the ability to rapidly explore chemical space and identify top-performing designs. Much of this, we observe, has to do with the aforementioned assumption of sufficiently small linkage chemistries being violated. For example, the inter-carboxyl benzene-ring-based chemistry shown in Fig. 10 E) is large enough to interact with atoms of the peptide chain, which is capped by the tetrahedral vertices. Unfortunately, we are unaware of an elegant solution to this problem; the addition of virtual atoms to a model's chemical graph is likely to cause significant issues, since each network has only learned a singular topology. Furthermore, only one universally transferable Boltzmann generator exists (Tan et al., 2025b), and it is trained on exclusively linear trajectories. As such, we adhere to the previously described approach: generate many samples and eliminate the problematic ones. Clashes are also observed, however less frequently, in regions far from the linkage since conditioning drives latents somewhat out of distribution by definition.

### B.1.2 CHEMISTRY-RESTRICTED CYCLIZATION ANALYSIS

The CYCLOPS framework trivially enables specific cyclization chemistries to be removed from consideration, which is useful when a particular cyclization is difficult to form or is prohibited. Yet it also yields insights into the effect of $P(C_i|x)$ (our exponential reweighting) on our loss minimization. Intuitively, the fewer allowed cyclizations, the greater the post-conditioning losses should be; fewer considered cyclizations means fewer options for the framework when annealing a given prior sample. As such, we condition DNF generation of the peptide chains used in Fig. 4, albeit restricted to just head-to-tail amide bonds, as this cyclization is common to all peptides. This provides a useful sanity check: if this distribution of H2T-restricted $\mathcal{L}_{cyc}$ is generally greater than that of all cyclizations, this suggests our CYCLOPS annealing protocol may correspond to valid cyclizations and that post annealing distributions of $\mathcal{L}_{cyc}$ may reveal how amenable a peptide is to the considered cyclizations. If, however, the underlying Boltzmann generator is not trained well, such that it produces structures with little correspondence to valid conformations, we would expect these distributions to be somewhat similar, as optima will be unrestrained by what is physically possible. Fortunately, the expected distinction between restricted and unrestricted cyclization is observed in Fig. 12. For all tested peptides, restricting loss minimization to just H2T amide bonding significantly increases the mean of the distribution and significantly alters the distribution itself under a two-sample Kolmogorov-Smirnov test.

### B.1.3 OPTIMIZATION DYNAMICS AND CONVERGENCE

While ablation and simulated annealing hyperparameter studies are left to future work, we do include a brief examination of the latent dynamics during the optimization process below. Fig. 13

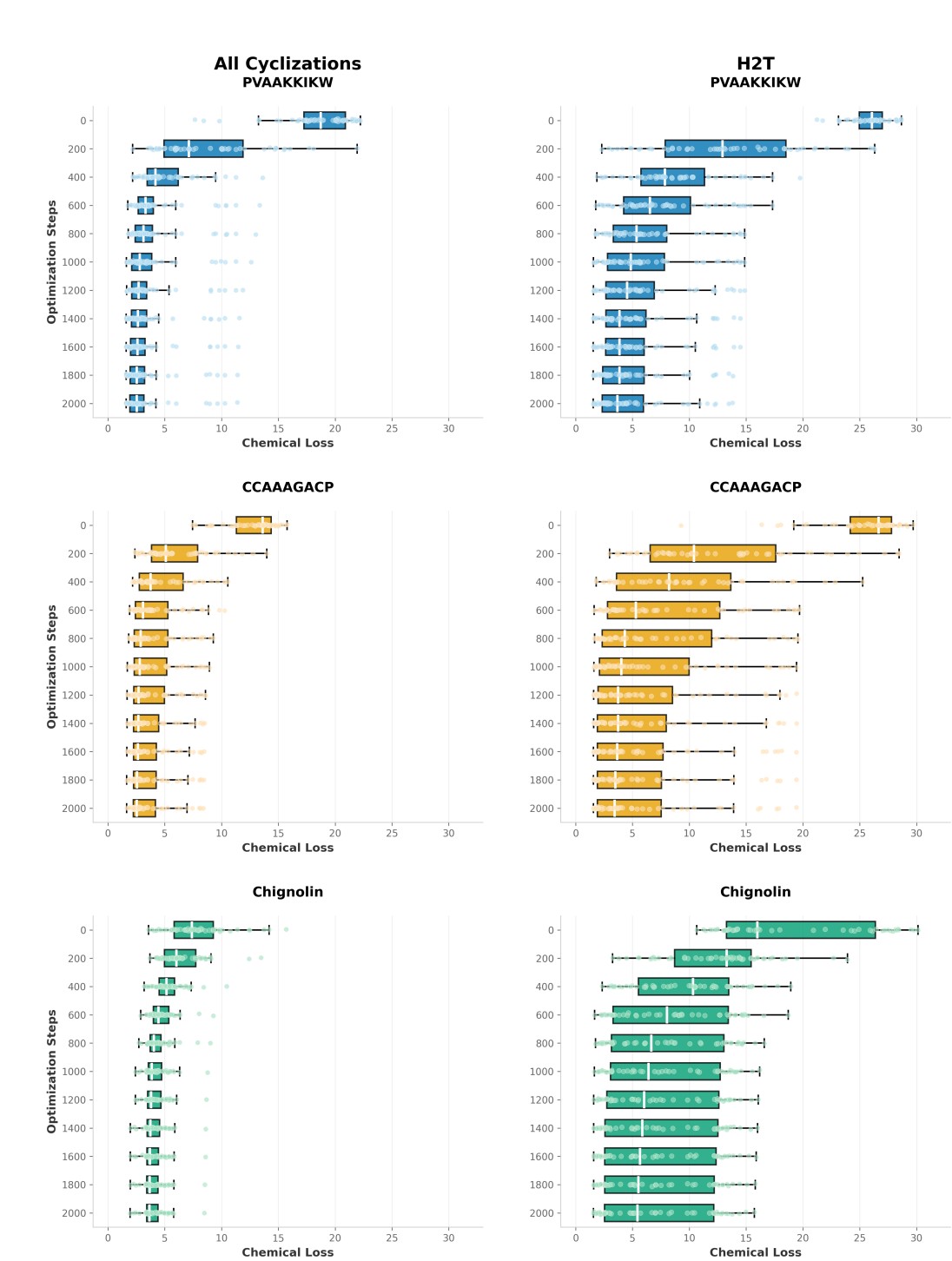

Figure 13: Box and whisker plots of CYCLOPS cyclic loss over 2000 steps of simulated annealing for peptide sequences from Fig. 4(rows) for all cyclizations (left column) and just head-to-tail (right column, H2T).

displays the progression of $\mathcal{L}_{\text{cyc}}$ values throughout the 2000-step simulated annealing process for both unrestricted and head-to-tail restricted cyclizations. Notably, the loss values appear to plateau in the latter stages of optimization. This either suggests that the simulation may be (1) converging rather quickly, and hence cooling too fast, or (2) that more steps than necessary are used during the optimization. Given Fig. 7 and Fig. 6, however, we suspect latent space is at least somewhat well explored. Still, this suggests the need of thorough studies on the effects of the various simulated annealing parameters on convergence and cyclization choice, which we leave for future work.

### B.2 CONDITIONAL NORMALIZING FLOW (ECNF++) FLOW GUIDANCE

We generally observe that, for the peptides tested, ECNF++ guided flow produces less satisfactory conformational samples than SBG simulated annealing. Of particular significance is that the peptides employed in this study are limited to those permitting only head-to-tail amide bond cyclization. This constraint arises from the scarcity of suitable MD trajectories in the existing literature—a limitation that is further exacerbated by the exponential scaling of inference time with system size for state-of-the-art CNF-based Boltzmann generators (Tan et al., 2025a). Consequently, only small peptides are currently computationally feasible, which severely restricts the already limited pool of available training data. Moreover, as loss guidance is inherently based on gradient descent, we suspect it may be more prone to getting trapped in local minima than simulated annealing, as it lacks the explicit exploratory potential of the latter.

Since the ECNF++ architecture is E(3) equivariant, it is prone to producing samples of incorrect chirality or mixed correct-incorrect chirality. Samples of entirely wrong chirality are mirrored and included, whilst those of mixed chirality are discarded. For Fig. 8, we begin with 128 unconditional and conditional samples and filter based on the aforementioned procedure.

## C LLM USAGE STATEMENT

Large language models (LLMs) were used to assist with writing (e.g., polishing phrasing, proofreading) and with code implementation support (e.g., debugging syntax and boilerplate generation). All outputs were reviewed and verified by the authors. LLMs were not used in research ideation or experimental design. The authors take full responsibility for the accuracy and integrity of the manuscript.

