# OpenReview forum: "Model Agnostic Conditioning of Boltzmann Generators for Peptide Cyclization"
_ICLR.cc/2026/Conference — Submitted to ICLR 2026_

### Official Review · Reviewer_T12j · 2025-10-14

**Soundness:** 4
**Presentation:** 4
**Contribution:** 4
**Rating:** 4
**Confidence:** 2

**Summary:**

This paper introduces CYCLOPS, a model-agnostic conditioning framework that enables Boltzmann Generators (both discrete and continuous normalizing flows) to generate cyclic peptide conformations without retraining. The method reformulates peptide cyclization as a conditional sampling problem, using tetrahedral geometric constraints derived from toy molecular dynamics simulations. It supports 18 crosslink chemistries and is validated on both SBG (DNF) and ECNF++ (CNF) architectures. The experiments demonstrate that CYCLOPS effectively biases latent distributions toward cyclic structures, offering a computationally efficient approach to macrocyclic peptide modeling.

**Strengths:**

CYCLOPS is conceptually novel, proposing a training-free conditioning approach that elegantly integrates chemical priors into generative models. The formulation is physically grounded, combining kernel density estimation with tetrahedral geometry constraints to model feasible cyclizations. The framework’s model-agnostic design allows application to diverse architectures, making it broadly useful for the peptide design community. Experimental validations on both DNFs and CNFs show improved cyclicity metrics and promising generalization across peptide sequences and chemistries.

**Weaknesses:**

Despite its originality, the study suffers from limited validation depth. The evaluation primarily focuses on geometric plausibility (CYCLOPS loss) rather than energetic stability or experimental fidelity. The assumption of “small linkage chemistry” constrains the generality, as larger linkers cause atom clashes and unrealistic structures. The lack of public code/data and incomplete analysis of computational efficiency limit reproducibility. Moreover, comparisons with recent diffusion-based molecular design frameworks are missing, and the physical realism of generated conformations is not systematically benchmarked.

**Questions:**

1. The paper relies mainly on internal loss functions; no physical or energy-based validation (e.g., RMSD to MD references, binding energy, or experimental data) is provided.
2. The “small linkage” assumption and fixed tetrahedral approximation may not generalize to complex or bulky cyclizations.
3. Many generated conformations (∼90%) contain steric clashes or unrealistic bond geometries, suggesting insufficient physical constraints.
4. Missing comparisons with diffusion or flow-matching models (e.g., DiffusionFlow, DiffDock, or recent peptide diffusion models).

---

### Official Review · Reviewer_6gdM · 2025-10-30

**Soundness:** 1
**Presentation:** 1
**Contribution:** 1
**Rating:** 2
**Confidence:** 5

**Summary:**

The paper proposes CYCLOPS, a model-agnostic framework that conditions Boltzmann generators to sample cyclic peptide conformations without retraining. Concretely, CYCLOPS defines a cyclicity loss via KDE fitted to toy MD simulations, and applies this loss to bias either DNF or CNF generators. Experiments with SBG and ECNF++ variants show reduced cyclic loss and illustrative cyclic structures.

**Strengths:**

1. The proposed loss can be plugged into both DNF and CNF Boltzmann generators, avoiding retraining and demonstrating cross-architecture applicability.
2. The paper reports multiple evidences supporting the resulting distribution shift after conditioning.

**Weaknesses:**

1. The method is presented vaguely, and the concrete integration strategies for both CNF and DNF remain under-specified in the main paper. For DNFs, the simulated annealing routine is mentioned only at a high level (Line 200), while for CNFs, a heuristic derivation is provided only in Appendix A.3. The paper should explicitly formalize how the cyclic loss is coupled to CNF vector fields and DNF latents within the main method section, including mathematical formulations and implementation details.

2. Direct comparisons with prior work are missed. The paper explicitly cites CPComposer [A], which addresses essentially the same task, that is to generate cyclic peptides from linear peptide data under composable geometric conditions. Given the strong task overlap and the centrality of conditioning for cyclization, a direct comparison is necessary to contextualize this paper’s contributions and to assess its actual performance improvements over existing approaches.

3. The overall structure of the paper is poorly balanced. The main text is under eight pages, while Figures 5–9 occupy a large portion but contain limited quantitative information. In contrast, the method section spans less than one page and lacks essential algorithmic detail. The authors should redistribute content, reducing excessive figure space and incorporating more methodological descriptions into the main text.

4. The trade-off of the guidance is not clearly discussed. Typically, stronger guidance would lead to better conditional satisfaction but lower distributional fidelity. This trade-off should be analyzed and validated through controlled experiments.

[A] Jiang, Dapeng, et al. "Zero-Shot Cyclic Peptide Design via Composable Geometric Constraints."

**Questions:**

See Weaknesses

---

### Official Review · Reviewer_jRg1 · 2025-10-31

**Soundness:** 3
**Presentation:** 2
**Contribution:** 2
**Rating:** 4
**Confidence:** 4

**Summary:**

This paper introduces **CYCLOPS** — a *model-agnostic conditioning framework* for Boltzmann Generators (BGs), enabling **cyclic peptide generation** without retraining.
The method reformulates cyclization as a *conditional sampling problem* using chemically informed **tetrahedral constraints**, derived from molecular dynamics (MD) simulations of small “toy” linkages.

Specifically:
1. A **Cyclic Loss (L_cyc)** encodes cyclization feasibility via kernel density estimation (KDE) over six interatomic distances defining a tetrahedron.
2. Conditioning can be applied post-hoc to *any* BG model — both discrete (DNF/SBG) and continuous (CNF/ECNF++) — using **latent-space simulated annealing** or **flow guidance**.
3. The framework is validated across several peptides (e.g., PVAAKKIKW, Chignolin) and 18 possible linkage chemistries.

CYCLOPS demonstrates that Boltzmann generators trained only on linear peptides can be reweighted to favor cyclic conformations through geometry-aware conditioning.

**Strengths:**

1. Reformulating peptide cyclization as *conditional reweighting* over linear conformational ensembles is conceptually creative and potentially impactful.
2. The framework is compatible with diverse Boltzmann generator architectures (DNF, CNF), aligning with recent work on flow-based generative modeling.

**Weaknesses:**

1. **Lack of Quantitative Validation**
   - The paper does not provide clear *quantitative metrics* for success beyond qualitative figures and loss distributions.
   - No RMSD-based evaluation, structure validity statistics, or benchmark comparisons to diffusion-based peptide design methods (e.g., Jiang et al., 2025; Zhou et al., 2025).
   - While Fig. 11–12 show loss reductions, it is unclear whether this correlates with *chemically correct* cyclizations.

2. **Over-Reliance on Toy Models**
   - The tetrahedral KDE approach is based on *small model linkages* that neglect larger steric effects or solvent-exposed flexibility.
   - The assumption that a four-atom tetrahedron adequately captures realistic cyclization geometry is debatable.
   - The paper itself acknowledges that about 10% of generated samples are actually valid (Sec. B.1.1).

3. **Empirical Weakness**
   - Despite claiming model-agnostic generality, ECNF++ conditioning produces visibly distorted structures and out-of-distribution conformations (Fig. 8–9).
   - No molecular energy validation (e.g., force-field evaluation, OpenMM minimization) is provided to confirm physical plausibility.

4. **Presentation and Clarity**
   - The exposition is mathematically dense and occasionally repetitive; core algorithmic insights (e.g., Fig. 3–5) are buried in appendices.
   - Figures lack clear legends or quantifiable axes, reducing interpretability.

**Questions:**

1. Can the authors provide quantitative comparisons (e.g., RMSD, dihedral recovery) between CYCLOPS-generated and reference cyclic conformations?
2. How sensitive are results to the KDE bandwidth (0.64 Å × I) or α in Eq. (5)?
3. Could the tetrahedral KDE model be extended with more atoms or angles to capture larger crosslink chemistry?
4. Have the authors attempted any *energy minimization or MD refinement* to verify physical plausibility?
5. How robust is the conditioning process when applied to unseen peptide sequences or longer chains?

---

### Official Review · Reviewer_3Gn5 · 2025-10-31

**Soundness:** 1
**Presentation:** 2
**Contribution:** 1
**Rating:** 2
**Confidence:** 2

**Summary:**

This paper introduces cyclOPS, a framework that takes an already trained Boltzmann generator on linear peptides, conditions it by adding cyclization sites (i.e. additional cyclization atoms ), and then generates conformations of the newly obtained cyclic peptide using temperature annealed sampling.

**Strengths:**

I didn't really understand the method of this paper, sorry. As such, it is hard to elaborate for me in this part.

**Weaknesses:**

Lack of clarity in the paper:

- for the methodology:
I am sorry, but I could hardly wrap my head around how the cyclization works and in the end I don't understand it. I do not understand the introduction of the tetrahedron linkage, how do you choose the linkage for each linear peptide, what is the simulated toy model and why we need to simulation, nor why KDE is used and why it is useful to generate conformations. Figure 3 does not help to understand. Could you please clarify this process in a rigorous way?

-  for the experimental results:
What is the goal of the figures shown in the experiments? We see that the modified 3D structures contain cycles, but there is no information about the quality of the generated structures in terms of how  Boltzmann energy, diversity of the generated conformers etc

Lack of formal definitions:
- What is a bound state?
- What is the definition of the chemical loss (used often but not defined ! )

Lack of mathematical & scientific rigor:
- "Intuitively nearly cyclic conformations ... amenable to a given cyclization " . Why is this? Do you have a reference for this?
- I do not understand the cyclic loss. How do you condition x with C_i , (i.e. how do you generate a conformation of x that contains a cycle C_i) ? Also, I thought the point was to generate cyclic peptides without retraining, so what is the goal of having a cyclic loss here ? How do you use it?

**Questions:**

(See the section above)

---

### Meta-Review · Area_Chair_CV5J · 2026-01-04

**Summary:**

Multiple reviewers highlight the importance of the application and the strengths of the proposed method, including its training-free nature. However, all reviewers raise two main concerns: poor presentation of the proposed method and insufficient validation. Although this might partly be due to the specific applications in chemistry considered in the paper, an ICLR publication must clearly present the proposed methodology.

**Reviewer Concerns:**

The authors have not engaged during the rebuttal period.

**Reviewer Scores:**

Due to the lack of the authors' response, I assume the scores could only decrease.

---

### Decision · Program_Chairs · 2026-01-26

Reject